# Integrative Single-Cell and Bulk RNA Sequencing Identifies a Glycolysis-Related Prognostic Signature for Predicting Prognosis in Pancreatic Cancer

**DOI:** 10.3390/ijms26115105

**Published:** 2025-05-26

**Authors:** Nan Wu, Chong Zhou, Xu Yan, Ziang Liu, Ruohan Jiang, Yuzhou Luo, Ping Jiang, Yu Mu, Shan Xiao, Xien Huang, Yunzhen Zhou, Donglin Sun, Yan Jin

**Affiliations:** 1Laboratory of Medical Genetics, Harbin Medical University, #157 Bao Jian Road, Harbin 150081, China; 2Key Laboratory of Preservation of Human Genetics Resources and Disease Control in China (Harbin Medical University), Ministry of Education, Harbin 150081, China; 3State Key Laboratory of Frigid Zone Cardiovascular Diseases, Harbin Medical University, Harbin 150081, China

**Keywords:** pancreatic cancer, pancreatic adenocarcinoma, glycolysis, risk model, ENO1, PGM2L1

## Abstract

Alterations in glycolysis play a crucial role in cancer cells, influencing tumor aggressiveness and therapeutic effect, particularly in pancreatic adenocarcinoma (PAAD). However, the specific glycolysis-related genes involved in PAAD progression remain poorly understood. This study established glycolysis-related molecular subtypes with distinct survival outcomes using TCGA datasets. The favorable prognosis subtype exhibited enhanced immune infiltration and an activated tumor microenvironment. A glycolysis prognostic model effectively predicted PAAD survival, correlating with global glycolytic pathways, and AUCell evaluated neutrophil communication networks of models. Functional validation demonstrated that ENO1/PGM2L1 co-expression promoted tumor proliferation, migration, invasion, and glycolytic flux in vitro, while accelerating xenograft growth in vivo. Conversely, their knockdown suppressed malignancy. Our study demonstrated that the glycolytic prognostic risk model serves as a reliable tool for prognosis and prediction of PAAD progression. ENO1 and PGM2L1 emerge as key risk factors promoting the malignant progression of PAAD.

## 1. Introduction

Pancreatic adenocarcinoma (PAAD) shows the poorest clinical outcome among all common solid cancers, with a five-year overall survival (OS) of merely 11% [1]. Despite the existence of standard treatments, PAAD remains the 3rd primary cancer-related mortality in the USA and 6th primary cause in China [2,3]. Projections indicate that PAAD will ascend to become the second principal reason for cancer deaths by 2030 [4]. Approximately 95% of PAAD cases manifest as exocrine cell tumors, with pancreatic ductal adenocarcinoma (PDAC) representing the predominant subtype. While advances in genomic research have contributed to improving the diagnostic and therapeutic strategies for PAAD, substantial progress remains elusive owing to the complex pathogenesis and molecular heterogeneity of the disease. Consequently, comprehensive clinical trials are urgently needed to elucidate the pathogenesis of PAAD and identify novel biomarkers that can predict prognosis and therapeutic effects.

In 1920, Otto Warburg first demonstrated that solid tumor cells exhibit higher glucose consumption compared to normal tissues, even under aerobic conditions. These tumor tissues often metabolize glucose-derived carbon into lactic acid via aerobic glycolysis, a well-documented metabolic phenotype in oncology [5]. Notably, PAAD cells display pronounced enhancement in glycolysis. This metabolic alteration is attributed to various factors, including oncogenes, key transcription factors, the hypoxic tumor microenvironment, and supportive stromal components such as cancer-associated fibroblasts and macrophages [6,7,8]. In fact, although glycolysis produces less energy compared to oxidative phosphorylation, it facilitates the production of metabolites crucial for the biosynthetic processes. This metabolic adaptation has an important function in facilitating cancer cell proliferation and enhancing survival. Consequently, PAAD displays heightened glucose uptake to fulfill its increased energy and biosynthetic demands, thereby offsetting the low efficiencies of energy production. Elevated blood sugar levels may exacerbate the malignant progression of PAAD by providing abundant glucose, thereby increasing the risk of poor prognosis. For instance, a meta-analysis demonstrated that for every 0.56 mmol/L increment in fasting blood glucose, there is a 14% rise in the incidence of PAAD [9].

Glycolytic enzymes are pivotal in driving tumor cell growth dominance via the Warburg effect during solid tumor progression. These enzymes not only facilitate anaerobic glycolysis of tumor cells but also exhibit moonlight functions [10]. Recent studies have demonstrated that glycolytic enzymes participate in regulating tumor metastasis, angiogenesis, and drug resistance. Furthermore, they affect the tumor immune microenvironment by modulating immune responses and promoting the expression of immune-related genes [8,11,12,13,14]. However, glycolytic enzymes’ utility as standalone prognostic and predictive biomarkers for PAAD remains largely uninvestigated.

Given the important role of glycolytic reprogramming in PAAD progression and lack of glycolysis-specific molecular subtype, this study aimed to identify molecular subtypes of PAAD by clustering glycolysis-related genes. Unlike previous analyses of individual glycolysis genes, the novelty of this study is identifying unique glycolysis molecular clusters that exhibit distinct immune infiltration and survival outcomes. Meanwhile, our research develops a prognosis risk model depending on glycolysis-related genes. The prognosis risk model is superior to the traditional models that rely on single-gene markers. This study deepens the understanding of how glycolytic dysregulation orchestrates tumor progression and provides a translational foundation for glycolysis-targeted therapy. Overall, this study fills the gap of glycolysis-related genes as prognostic targets for PAAD treatment, positioning glycolysis not only as a metabolic marker but also as a feasible target for subtype and prognosis. Figure 1 demonstrates the main content of this research.

## 2. Results

### 2.1. Identification of Two Molecular Subtypes of PAAD Depending on Glycolysis-Related Genes

Clinical and transcriptome data of 182 PAAD samples were obtained from the TCGA database. A curated set of 92 glycolysis-associated genes (GSEA criteria) was acquired from Reactome. Univariable Cox analysis generated 25 glycolysis-related prognostic genes, and the consensus clustering divided the PAAD samples into two subgroups based on these genes (Appendix A). The optimal clustering was identified when k = 2 (Figure 2A–C). A total of 91 patients were clustered into cluster 1, and another 91 patients were clustered into cluster 2. Survival curves found that patients assigned to cluster 2 exhibited a significantly extended OS (*p* = 0.024; Figure 2D).

We conducted investigations into the disparity in immune infiltration levels across the two clusters. Using “ESTIMATE v1.0.13”, we found cluster 2 displayed elevated immune scores (*p* = 0.000072), ESTIMATE scores (*p* = 0.00011), and stromal scores (*p* = 0.027), along with lower tumor purity (*p* = 0.00011), when contrasted with those in cluster 1 (Figure 2E). “ssGSEA 1.50.0” showed infiltration degrees of multiple cells in cluster 2 are more abundant than in cluster 1, including activated B cells, activated CD8+ T cells, and so on (Figure 2F). The “CIBERSORT v0.1.0” algorithm achieved the same results (Appendix A). To evaluate the sensitivity difference between the two clusters to drug resistance, we employed the “oncopredict v0.2” algorithm to determine the semi-maximum inhibitory concentration (IC50) for each cluster. Despite differing molecular subtypes and prognoses within the clusters, we observed no significant differences in their drug responsiveness (Figure 2G). The genomic mutation analysis showed that cluster 1, with a poorer prognosis, has a higher burden of genomic mutations (Figure 2H).

These results demonstrate that PAAD patients were divided into two molecular subtypes based on glycolysis-related genes, which relate to significant variances in immune infiltration and genomic mutation between the two subtypes.

### 2.2. The Different Expressed Genes and Enrichment Analysis of Two Subtypes

To investigate fundamental distinctions in biological functions and signaling pathways between these two molecular subtypes, we performed transcriptome differentially expressed genes analysis. Among the 298 DEGs identified, 176 genes showed increased expression while 122 exhibited decreased expression in Cluster 1 (Figure 3A,B). KEGG enrichment analysis on DEGs identified many functional pathways related to the pancreas, including pancreatic secretion, protein digestion and absorption, and glycolysis/gluconeogenesis, indicating heterogeneity in pancreatic function between the two molecular subtypes (Figure 3C). Additionally, key biological processes related to tumorigenesis and metastasis, including the PI3K-AKT signaling pathway, showed significant enrichment. The molecular function of GO enrichment analysis is also enriched in the related functions of tumorigenesis and metastasis (Figure 3D). We also analyzed the biological processes and cellular components of GO analysis as presented in Appendix A. GSEA revealed that multiple glucose metabolism-related pathways, including glycolysis and the pentose phosphate pathway, were significantly enriched in Cluster 1 (Figure 3E,F).

These results indicated distinct disparities in pancreatic function, glucose metabolism, and tumor-related processes between the two glycolysis-associated molecular subgroups. These above results show that these glycolysis-related genes are not only involved in glycolysis but also further affect the development of tumors, further verifying the accuracy of our molecular subtypes.

### 2.3. Glycolysis-Related Gene’s Prognostic Risk Model Was Constructed by Machine Learning

Subsequently, the LASSO regression algorithm was used to construct a clinical prognostic model with these 25 glycolysis-associated genes. Approximately 25 glycolysis-related genes were subjected to univariate Cox regression analysis (Appendix A) to acquire 10 genes, and finally a prognostic risk model was constructed using three hub genes (Figure 4A–C). In this model, the risk score of each PAAD sample was computed by applying the subsequent formula: Risk Score = 0.606 × NUP37 + 0.202 × PGM2L1 + 0.202 × ENO1. The prognostic risk model effectively stratified PAAD samples into two subgroups, with the high-risk subgroup exhibiting significantly poorer overall survival (Figure 4D). The ROC curve proved the prognostic risk model has a good predictive value (Figure 4E, 1-year AUC = 0.73, 3-year AUC = 0.83, 5-year AUC = 0.89).

There were no significant differences in ages, genders, and metastasis statuses between the two groups (Appendix A), and the risk model still showed predictive performance when regrouped by age and gender (Appendix A). The multivariate COX regression of TCGA-PAAD further demonstrated that the risk score was recognized as a prognostic factor (Appendix A). Two external GEO datasets were employed to validate the model’s robustness (GES62452 and GSE78229 datasets, Figure 4F,G). ROC analysis measured the discrimination of glycolysis-related genes, with 1-, 3-, and 5-year AUCs of 0.69, 0.86, and 0.94 in GSE662452; 0.74, 0.89, and 0.92 in GSE78229; and 0.73, 0.83, and 0.89 in TCGA-PAAD, respectively (Figure 4H). Subsequently, the C-index values evaluation found that the predictive efficiency of the two GEO tests exhibited higher accuracy (GSE62452: 0.679; GSE78229: 0.650; TCGA-PAAD: 0.483, Figure 4I).

Meanwhile, the low-risk group is more sensitive to 5-Fluorouracil, Sorafenib, Cisplatin, and Dactolisib (Figure 4J). These findings corroborate our earlier observations: the high-risk group exhibited a higher burden of genomic mutations (Figure 4K). These findings indicated that the developed prognostic model can effectively predict outcomes in PAAD patients.

### 2.4. Identification of Highly Correlated Gene Modules in Risk Model

To measure gene correlation within the model group, we use “WGCNA v1.72-5” to construct gene co-expression networks (Figure 5A–C). By calculating the correlation matrix between Module Eigengenes, modules whose similarity is higher than a set threshold are merged (Figure 5D,E). The highest correlation was demonstrated by the “brown” module (R = 0.83) (Figure 5F,G). The prognostic model group was similar to the gene module of patient prognosis. KEGG enrichment analysis of the brown module revealed key pathways, including glycolysis and JAK-STAT signaling (Figure 5H). These results highlighted the significance of the model’s predictive accuracy for clinical outcomes and identifying high-risk patients through dysregulated glycolysis pathways.

### 2.5. Evaluation of Glycolysis Prognosis Model in Pancreatic Cancer Through scRNA-Seq Analysis

To evaluate the glycolysis prognostic model at single-cell resolution, we analyzed a corresponding dataset, GSE212966, with 6 pancreatic cancer specimens. Dimensionality reduction via PCA followed by t-SNE clustering resolved nine distinct cell populations. Through differential expression analysis across all cell clusters, they were marked as Epithelial cells, Fibroblasts, Mast cells, Endothelial cells, Antigen-Presenting Cells, Neutrophils, Pancreatic Stellate Cells, B cells, and T cells (Figure 6A). Visualize the characteristic genes of each type of cell (Figure 6B). Through single-cell localization visualization of the glycolytic prognostic model factors, the risk factor ENO1 exhibited the strongest expression (Figure 6C). We utilized AUCell to evaluate the activity of the gene set of the glycolysis prognosis model in each cell of the single-cell transcriptome (Figure 6D). Using the glycolysis gene set of GSEA as a control, we found that both the glycolysis prognosis model and the glycolysis gene set had relatively high AUCell values (Figure 6E,F). We also found that the cell subtypes with a high AUCell score of the glycolysis prognosis model were mainly concentrated in Neutrophils (Figure 6G,H). CellphoneDB analysis demonstrated Neutrophils among the AUCell glycolysis high-activity cells exhibited stronger cell communication with B cells, T cells, and other cell types (Figure 6I,J).

### 2.6. PGM2L1 and ENO1 Promote Proliferation, Migration, Invasion, and Glycolysis of PAAD

To assess the prognostic model’s reliability, we investigated the individual roles of each risk factor gene in PAAD. GIEPIA (http://gepia2.cancer-pku.cn/, accessed on 6 May 2025) [15] and UALCAN (https://ualcan.path.uab.edu/, accessed on 5 March 2025) [16] analysis platforms showed that the abundance of mRNA and protein associated with NUP37, PGM2L1, and ENO1 were higher in PAAD than in normal pancreatic tissue, and upregulated expression of individual genes predicted worse clinical outcomes (Appendix A). The Timer 2.0 platform also indicated that all three single genes had immune infiltration with poor prognosis (Appendix A) [17]. Since the protein expression differences of PGM2L1 and ENO1 are more significant, we further investigate their function in PAAD through in vitro and in vivo experiments.

According to endogenous expression, we selected the PAAD cell line. MIA Paca-2 was used to establish a stable overexpression cell model, overexpressing PGM2L1, ENO1, or both co-expression (Figure 7A,B). PGM2L1 and ENO1 were silenced by short hairpin RNA (shRNA) in Panc-1 and MIA Paca-2, respectively (Figure 7C). Cell proliferation and colony formation assays showed that cells overexpressing PGM2L1 or ENO1 significantly enhanced the proliferation ability, while cells co-expressing PGM2L1 and ENO1 exhibited the strongest proliferative ability, and knockdown of PGM2L1 and ENO1 reduced cell growth and colony formation (Figure 7D–F). Migration and invasion assays demonstrated that exogenous overexpression of PGM2L1 or ENO1 significantly enhanced cell motility (Figure 7G) and invasion potential (Figure 7H). Consistent with cell growth assays, cells co-expressing PGM2L1 and ENO1 showed the strongest motility and invasion potential. Knockdown of PGM2L1 and ENO1 significantly impaired the migratory and invasive capacities of cells (Figure 7I,J).

Since PGM2L1 and ENO1 were key enzymes involved in the glycolysis pathway, we investigated the changes in glycolysis further. Extracellular acidification rate (ECAR) was assessed to evaluate capacity and potential of glycolysis. The results demonstrated that exogenous overexpression of PGM2L1 and ENO1 enhanced cellular glycolytic capacity, and knockdown of PGM2L1 and ENO1 significantly decreased the glycolysis capacity of cells (Figure 7K). These results demonstrate ENO1 and PGM2L1 prove malignant progression of PAAD by promoting the proliferative, migratory, invasive capacities and glycolytic activity of PAAD cells. When PGM2L1 and ENO1 are co-expressed, these functions can be synergistically enhanced.

### 2.7. PGM2L1 and ENO1 Promoted Xenograft Tumor Growth in Mouse Models

We further explored the impact of PGM2L1 and ENO1 in vivo. Subcutaneous injection was used to generate mouse xenograft tumor models of MIA Paca-2 cells overexpressing PGM2L1, ENO1, co-expressing, and the control into the right dorsolateral regions of these nude mice (*n* = 4 per group). Tumor volume was monitored by caliper. After 32 days of tumor growth, the tumor tissues were dissected and weighed. The results showed that PGM2L1 and ENO1 promote the growth of xenograft tumors, and the tumors in the co-expression group were larger than other groups (*p* < 0.05; Figure 8A–C). IHC staining showed that the Ki67 level was increased in PGM2L1 and ENO1 co-expressing cells (Figure 8D). These findings show PGM2L1 and ENO1 accelerate tumor growth in xenograft models, supporting their oncogenic roles.

## 3. Discussion

As demonstrated in previous research, glycolytic PAAD is a subtype with a poorer prognosis. However, there is still a significant gap in identifying glycolytic markers that can make accurate prognostic predictions for PAAD patients [18]. We identified two distinct glycolysis molecular subtypes in PAAD, each associated with a different prognosis. In addition, we used machine learning to establish a refined prognostic risk model that provides greater accuracy in predicting outcomes. By combining in vitro and in vivo experiments, we demonstrated that the risk factors PGM2L1 and ENO1 play significant roles in modulating the malignant progression of PAAD cells.

Genes related to glycolysis are essential for the malignant development of PAAD. Therefore, characterizing the glycolysis-related phenotypes in PAAD will promote our understanding of the impact of glycolysis on its progression and contribute to the development of new prognostic markers. In this study, we applied single-variable Cox regression analysis to filter single genes associated with glycolysis that impact the prognosis of PAAD. Subsequently, 25 glycolysis-related genes were identified and analyzed, resulting in two distinct subgroups exhibiting unique glucose metabolism characteristics. Due to accumulating evidence indicating that tumor metabolism has a broader influence on regulating anti-tumor immune responses, further research is needed to investigate its prognostic significance in PAAD [19].

Chronic inflammation shapes the immunosuppressive microenvironment and becomes a core pathological driver for the malignant progression of PAAD [20]. Research has shown that the infiltration level of immune cells can be a reliable indicator of the microenvironment in PAAD [21]. ESTIMATE, CIBERSORT, and ssGSEA tools provide accurate quantification of immune and stromal cells contained in the cancer microenvironment, and we evaluate immune infiltration in the two molecular subtypes using these three methods. Consistent with prior findings, our data reveal that a low immune infiltration score and immunosuppressed status may indicate a poorer prognosis.

Recent studies have shown that pancreatic juice pathology and molecular analysis are crucial for diagnosing early PAAD, and a hyperglycemic tumor microenvironment helps accelerate the progression of PAAD [22,23]. Our enrichment results consistently demonstrate a correlation with the observed variations in pancreatic secretion, glycolysis, gluconeogenesis, and pentose phosphate pathways. We also enriched the PI3K-AKT signaling pathway, which is strongly associated with malignant phenotypes of PAAD [24,25,26]. The results indicate glycolysis-related genes are essential for pancreatic function, and any interference in glucose metabolism may synergistically promote the activation of tumor-related signaling pathways to promote tumor development.

In the past few years, machine learning algorithms have been used to discover potential associations between omics data and disease, thereby creating predictive models. LASSO regression analysis provides a good predictive model for screening significant variables associated with prognosis. Based on the previous results, we further established a glycolysis-related PAAD prognostic risk model through LASSO. NUP37, PGM2L1, and ENO1 were identified as prognostic risk factors, and they were pairwise correlated in PAAD. After analyzing the mRNA and protein expression levels of three genes in PAAD, we carried out in vitro and in vivo investigations on PGM2L1 and ENO1.

Phosphoglucomutases (PGMs) are an important group of enzymes widely present in human tissues. PGMs can catalyze the mutual conversion of glucose-1-phosphate and glucose-6-phosphate, which plays an important role in sugar metabolism. Our research finds the high expression of PGM2L1 in PAAD is beneficial for promoting tumor malignant progression and glycolysis. PGM2L1 acts as an indicator of prognosis, signifying a higher likelihood of adverse outcomes in PAAD.

ENO1 is one of the key enzymes in glycolysis and an oncogenic protein with diverse functions, overexpressed in most human cancers [27]. ENO1 is involved in promoting tumor progression by enhancing various cellular functions. It not only speeds up glycolysis but also promotes cancer cell proliferation and invasion, induces drug resistance, and activates oncogenic signaling pathways, including PI3K-AKT, HGFR, and WNT signaling pathways [28,29]. These results suggest that the risk factors ENO1 and PGM2L1 can synergistically promote malignant biological behaviors such as proliferation, migration, and glycolysis. These results also indicate that our established prognostic risk model has a certain reference value for evaluating the prognosis of PAAD patients.

Although no inhibitors of PGM2L1 have been discovered, inhibitors of ENO1 have shown good tumor suppression effects. Acting as a small molecule enolase inhibitor, POMHEX has the ability to selectively eliminate ENO1-deficient glioma cells at low nanomolar concentrations [30]. POMHEX can also work with anti-angiogenic drugs to effectively eliminate the drug-resistant tumors in the mouse model of colorectal cancer [31]. SF2312 is a phosphonate antibiotic. Through structural analysis and experiments, it has been proven to be an efficient nanomolar-level enolase inhibitor that has potential value in the treatment of cancers with ENO1 deficiency [32]. In addition, the cancer-specific functions of glycolytic enzymes and the therapeutic potential of targeted drugs have shown promising prospects for clinical transformation [33]. It is expected that more drugs targeting glycolysis will be applied in clinical practice in the future.

Several limitations are inherent in this study. The animal model we selected was a subcutaneous xenograft tumor model of thymus-free mice. Animal models cannot replicate the microenvironment of PAAD. And the prognosis model is based on TCGA databases; we will combine clinical samples to further verify the applicability of the model. Meanwhile, even though this research uncovers the significant roles that ENO1 and PGM2L1 play in glycolysis and the advancement of cancer, the specific molecular mechanism they regulate and the interaction mechanisms with the immune microenvironment require further elucidation in subsequent research.

In summary, our study uses consensus clustering to describe two different glycolysis-related gene-related molecular subtypes in PAAD. The two subtypes have differences in prognosis, immune infiltration, and drug resistance. By further establishing a risk prognosis model, we found that glycolysis-related genes can accurately predict the prognosis, and glycolytic heterogeneity significantly affects the malignant phenotype of PAAD. The combined expression of risk factors ENO1 and PGM2L1 was demonstrated for the first time to significantly synergistically promote the malignant progression of PAAD. Our research findings present new outlooks for the potential development of glycolysis-related targeted therapies for PAAD and new ideas for novel tumor markers that reflect the malignant progression of pancreatic cancer.

## 4. Materials and Methods

### 4.1. Data Acquisition

FPKM, gene expression counts, and clinical data of the TCGA-PAAD were downloaded from the UCSC Xena (https://xenabrowser.net/datapages/, accessed on 15 May 2022). The GSE62452 and GSE78229 datasets contain 65 tumor samples and 49 tumor samples of PAAD patients, respectively, and were analyzed as validation models and were acquired from the GEO database (http://www.ncbi.nlm.nih.gov/geo/, accessed on 6 July 2024). Then, Ensembl IDs were transferred to official gene symbols R package “org.hs.eg.db 3.12.0”. A total of 92 glycolysis genes were downloaded from Reactome (https://reactome.org/, accessed on 21 May 2022).

### 4.2. Identification of Molecular Subgroups

Using the R packages “Survival 3.3-1” and “ConsensusClusterPlus 1.54.0”, we first identified 25 glycolysis-related genes associated with PAAD clinical outcome. Carried out consensus clustering on their expression matrix to define molecular subtypes.

### 4.3. Analysis of Immune Cell Infiltration

Three methods were used to assess different molecular subgroups’ state of immune cell infiltration. ESTIMATE is an algorithm for analyzing tumor cell purity and stromal cell abundance [34]. We receive the stromal and immune signature in different molecular subgroups using the R package ‘estimate 1.0.13’. Using CIBERSORT, the proportions of immune cells in two clusters were computed, and ssGSEA was applied to compute immune cell enrichment scores [35].

### 4.4. Identifying Differentially Expressed Genes and Functional Enrichment

With RNA-seq count data, the mRNA differential expression genes were obtained by the R package ‘limma 3.46.0’ [36]. In the differential gene expression analysis, the limma package uses a linear model approach. A moderated *t*-test was employed to calculate the *p*-values for each gene and accounts for the variance across samples. To address the issue of multiple comparisons, the Benjamini-Hochberg method was used to adjust the *p*-values. Then, we used the ‘clusterProfiler 3.18.1’ R package to explore the differential gene’s function [37].

### 4.5. Genomic Mutation and Drug Sensitivity Analysis

Using the “TCGAbiolinks 2.31.2” and “maftools 2.18.0”R packages, we downloaded the TCGA PAAD patient MAF file and generated a waterfall plot to visualize genomic mutation distributions across clusters and risk groups [38]. The oncopdict algorithm, which leverages transcriptome expression levels, was employed to predict drug sensitivity levels [39]. Gene expression profiles across various cell lines and corresponding semi-maximum inhibitory concentration (IC50) for each drug were obtained from the GDSC (https://www.cancerrxgene.org/, accessed on 21 August 2023) dataset. Subsequently, the IC50 values of each drug were analyzed within the two molecular subtype clusters and the high-low risk model. The sensitivity of the drugs between two molecular subtype clusters and the high-low risk model was assessed utilizing an unpaired *t*-test.

### 4.6. WGCNA

WGCNA was employed to build coexpression networks, detect gene modules linked to pancreatic cancer phenotypes, determine an optimal soft thresholding power for scale-free network topology, and associate modules with clinical characteristics to isolate the most pertinent module for subsequent analyses.

### 4.7. Single-Cell Sequencing Data Processing

‘Seurat v4.2.2’ was used for single-cell RNA analysis. After dataset integration, quality control steps filtered out cells with mitochondrial gene expression >10%, ribosomal gene expression >40%, or nFeature_RNA values outside the 500–2500 range. The SCTransform function normalized and scaled the data for downstream analysis. Using the first 15 principal components, RunUMAP performed dimensionality reduction and clustering to generate cell clusters. SingleR (Version: 2.4.0) annotated cell types by referencing published marker genes, and the FindAllMarkers function identified unique molecular markers for each cluster. DoubletFinder detected and excluded doublet cells to refine the dataset.

### 4.8. AUCell Scoring

To detect cell clusters influenced by the prognostic model and glycolytic pathways, this study applied AUCell analysis to single cells using gene signatures from the prognostic model and glycolysis-related gene sets.

### 4.9. CCK-8 and Colony Formation

Cell Counting Kit 8 assays (New Cell and Molecular Biotech, Newcastle upon Tyne, UK, C6005) measured cell proliferation: 2 × 10^3^ cells/well in 96-well plates were assayed for OD_450_ using a Tecan F50 reader (Kawasaki, Japan) every 24 h for 5 days.

Colony formation: 0.5 × 10^3^ cells/6-well were cultured for 14 days, rinsed with PBS, preserved in 4% paraformaldehyde, dyed with gentian violet (Solarbio, Beijing, China, G1063), imaged via ChemiDoc MP, (Bio-Rad, Hercules, CA, USA) and colony numbers quantified by ImageJ (Version: 1.8.0.112).

### 4.10. Transwell Assay

The Transwell migration and invasion assay used 24-well plates: 5 × 10^4^ cells in 2% FBS upper chambers and 20% FBS lower chambers as chemoattractant. After 24 h (migration)/48 h (invasion) at 37 °C, upper membrane cells were removed. Migrated/invasive cells were fixed in 4% formaldehyde, stained with H&E (5/2 min), and quantified via four random Leica microscope fields (Wetzlar, Germany).

### 4.11. Western Blot

Western blotting followed standard protocols: cellular protein was extracted with cold RIPA buffer (Thermo, Waltham, MA, USA, 89901) + 1% protease inhibitors (Bimake, Shanghai, China, B14001), separated by 10% SDS-PAGE, and conveyed to PVDF (Millipore, Burlington, MA, USA, ISEQ00010). PVDF was blocked with 5% BSA, then incubated with anti-ENO1 (1:1000, Proteintech, Rosemont, IL, USA), anti-PGM2L1 (1:1000, Proteintech), and anti-GAPDH (1:2000, Proteintech). After washing, HRP-secondary antibodies (CST, 7076/7074) were applied, and bands were visualized via NcmECL Ultra (New Cell, Newcastle upon Tyne, UK, P10300) and imaged on ChemiDoc MP (Bio-Rad, Hercules, CA, USA).

### 4.12. Glycolysis Stress Assay

The glycolytic function was measured using the Seahorse XFp Analyzer (Agilent, Santa Clara, CA, USA). Approximately 2 × 10^4^ cells/well were plated overnight, then switched to XF RPMI medium (2 mM glutamine) and pre-incubated in a CO_2_-free environment for 1 h. The XFp automatically injected glucose (10 mM), oligomycin (5 μM), and 2-DG (50 mM) to measure ECAR. Wave Pro software (Version: 2.6.3.5) analyzed glycolysis, glycolytic reserve, and non-glycolytic acidification.

### 4.13. Xenograft Formation Assay

Animal studies conformed to ethical norms and were permitted by the Harbin Medical University Ethics Committee (HMUIRB2022018). Approximately 4- to 5-week-old BALB/c-nude mice (Shanghai SLAC, Shanghai, China) were housed under SPF conditions. Xenografts were induced by injecting 8 × 10^6^ cells (PBS suspension) into the right flank. Tumor volumes were measured twice weekly using calipers (formula: TV = [L × W^2^]/2). After 4–5 weeks, tumors were resected, weighed, and imaged.

### 4.14. Immunohistochemistry (IHC) Analysis

Deparaffinized/rehydrated sections were processed with 3% H_2_O_2_ (10 min, RT). Primary antibodies (PGM2L1, ENO1, Ki67; 1:200) were applied overnight. Then incubated with biotinylated anti-rabbit IgG (30 min, 37 °C), washed three times, and developed using DAB.

### 4.15. Statistical Analysis

R (v4.0.5) and SPSS 21.0 performed statistical analyses. Student’s *t*-test/Mann–Whitney U test compared two groups. Cox regression (univariate/multivariate) analyzed survival data. Kaplan–Meier curves with log-rank tests assessed overall survival. Pearson’s correlation evaluated PGM2L1/ENO1/NUP37 relationships. Data meant SD; *p* < 0.05 was significant.

## 5. Conclusions

Two molecular subtypes of PAAD were discovered in this study using glycolysis-related genes, with distinct differences in OS, immune infiltration, and genomic mutations. A prognostic risk model incorporating NUP37, PGM2L1, and ENO1 effectively predicted patient outcomes, highlighting these genes’ roles in tumor progression. These findings boost the comprehension of PAAD heterogeneity and offer potential avenues for targeted therapies and improved patient stratification.

## Figures and Tables

**Figure 1 ijms-26-05105-f001:**
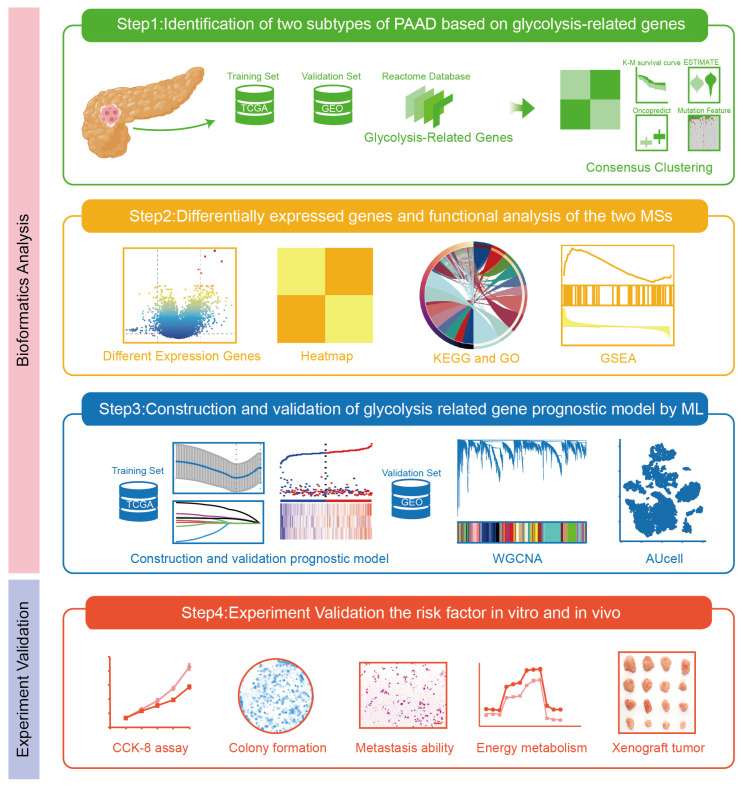
The schematic diagram about analysis workflow (magnification 200×, scalebar = 50 μm). TCGA, The Cancer Genome Atlas; GEO, Gene Expression Omnibus; MSs, molecular subtype; KEGG, Kyoto Encyclopedia of Genes and Genomes; GO, Gene Ontology; GSEA, Gene Set Enrichment Analysis; ML, machine learning; WGCNA, weighted correlation network analysis.

**Figure 2 ijms-26-05105-f002:**
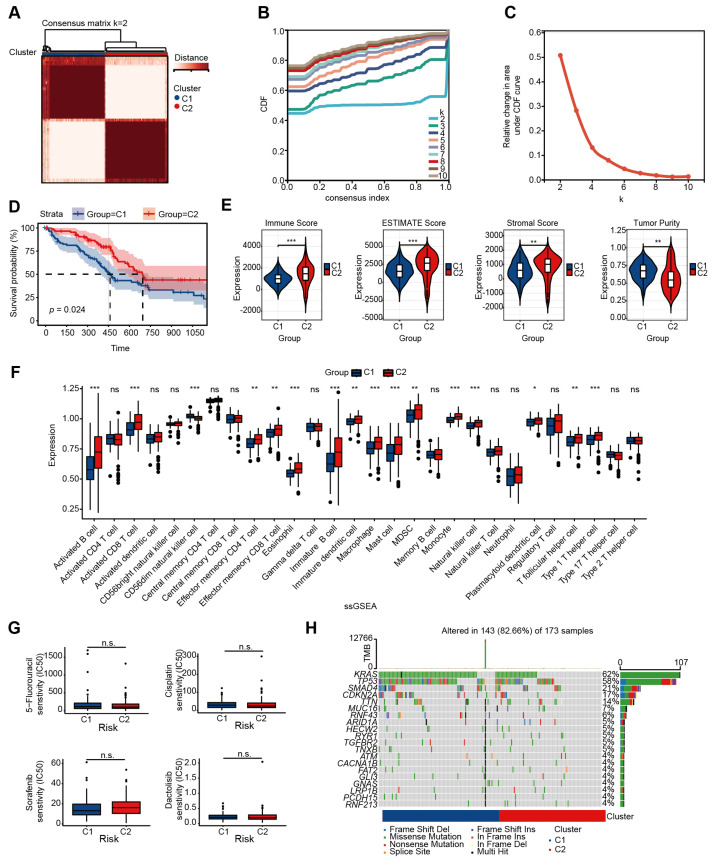
Consensus cluster and immune analysis in the two clustered subgroups. (**A**–**C**) Through consensus clustering, we found that two clusters (k = 2) provided the best solution. (**D**) Kaplan–Meier of the two clusters. (**E**) Immune score, ESTIMATE score, stromal score, and tumor purity calculated by ESTIMATE. (**F**) ssGSEA evaluated the expression of microenvironment cells. (**G**) Drug resistance scores were estimated by the “oncopredict” R package using GDSC data as the training cohort. (**H**) The somatic mutation features of the two clusters. (* *p* < 0.05; ** *p* < 0.01; *** *p* < 0.001, n.s. = no significance; independent Student’s *t*-test.).

**Figure 3 ijms-26-05105-f003:**
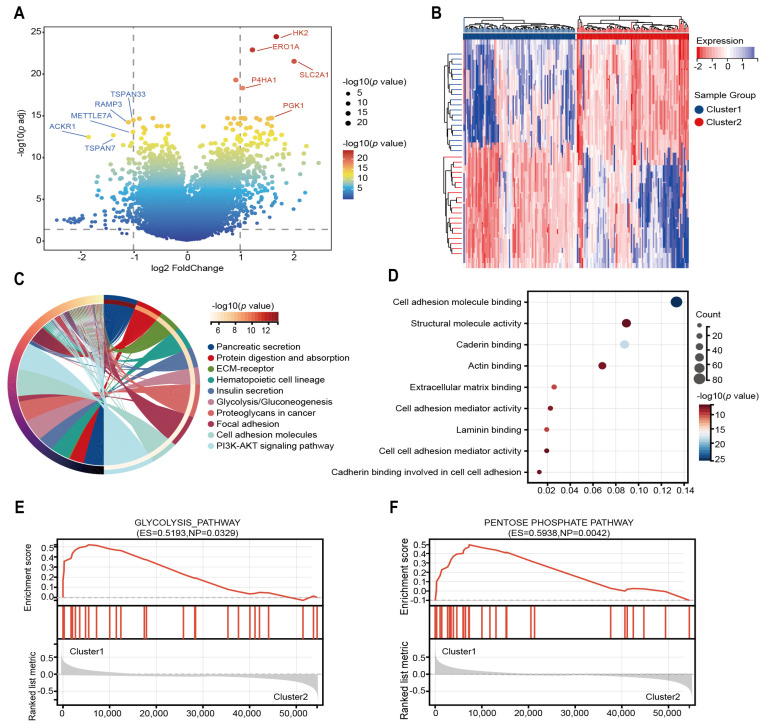
DEGs analysis and some functional analysis. (**A**) Volcano plot illustrating DEGs across two subgroups. (**B**) Heatmap illustrates the expression patterns of DEGs across subgroups. (**C**) Circular plot showing KEGG-enriched signaling pathways. (**D**) Bubble plot displaying GO-enriched biological processes. (**E**,**F**) GSEA plots presenting the enrichment results of pathway analyses.

**Figure 4 ijms-26-05105-f004:**
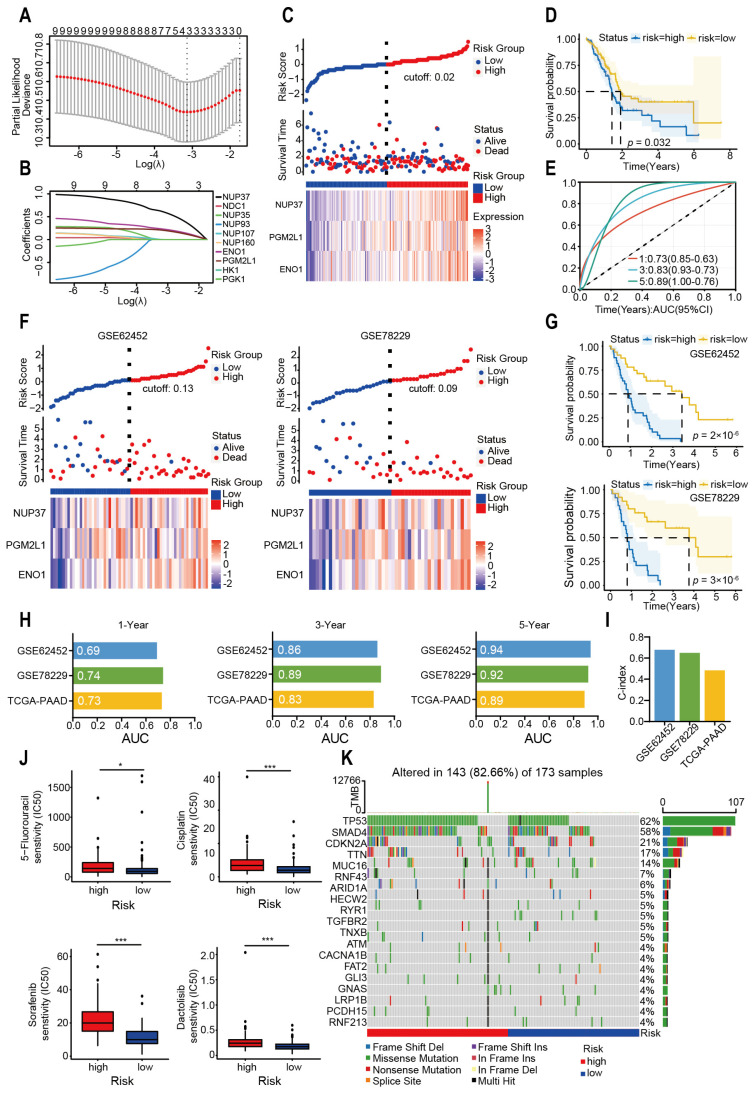
Development of a Machine Learning-Based Prognostic Risk Model. (**A**,**B**) LASSO regression analysis with optimal λ selection. (**C**) Visualization of survival time, risk score, and gene expression heatmap for PAAD patients. (**D**) Kaplan–Meier survival curves comparing the two risk subgroups. (**E**) Results about the ROC curve of the prognostic model are demonstrated. (**F**) Visualization of survival time, risk score, and gene expression heatmap in the two verification cohorts. (**G**) Kaplan–Meier survival curves in the two verification cohorts. (**H**) The AUC values of ROC curve for the training and test cohorts. (**I**) The C-index of train cohort and test cohorts. (**J**) Drug resistance scores were estimated by the “oncopredict” R package using GDSC data as the training cohort. (**K**) The somatic mutation features of the two risk groups. (* *p* < 0.05; *** *p* < 0.001, Wilcoxon rank-sum test.).

**Figure 5 ijms-26-05105-f005:**
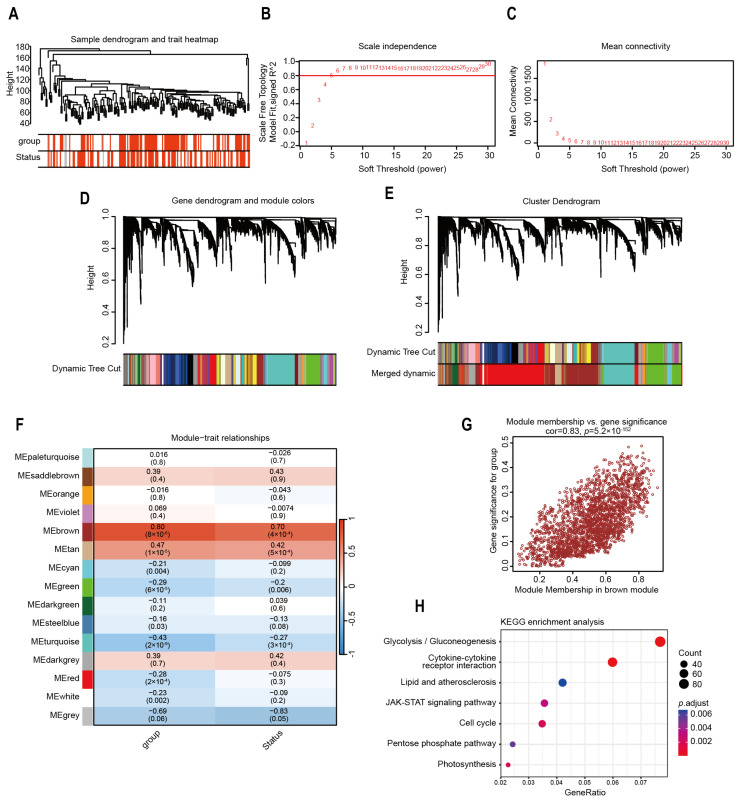
Recognition of highly correlated gene modules using WGCNA. (**A**) Using gene expression data, a dendrogram for hierarchical clustering was developed. (**B**,**C**) To construct scale-free networks, we performed the calculation of the soft thresholding parameter, depicting the scale-free fit index and mean connectivity across various power values. (**D**) Dynamic tree cutting and subsequent branch merging were employed to cluster modules into distinct groups. (**E**) Merged dynamic is used to shrink and integrate highly related modules to form the final set of optimized modules. (**F**) The average correlation of each module with risk group and prognosis status. (**G**) The relationship between membership in the brown module and gene significance was illustrated by a scatter plot. (**H**) Bubble plot depicting KEGG-enriched signaling pathways.

**Figure 6 ijms-26-05105-f006:**
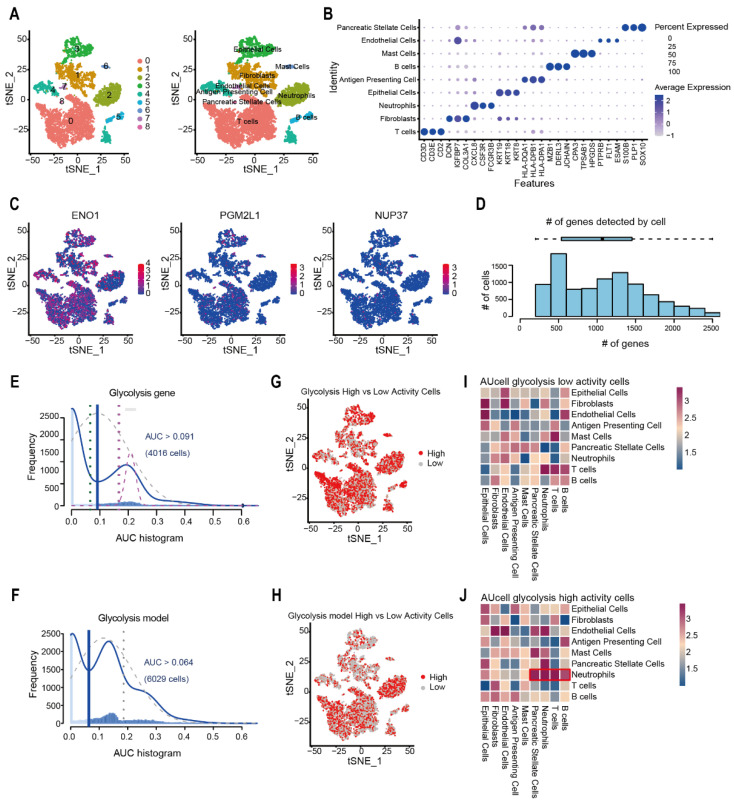
Evaluation of Glycolysis prognosis model in Pancreatic cancer Through scRNA-Seq Analysis. (**A**) t-SNE plot depicting 29 cell clusters and their corresponding cell types. (**B**) Cell types identified in the t-SNE plot were annotated based on marker genes. (**C**) t-SNE plots visualizing single-cell expression of ENO1, PGM2L1, and NUP37. (**D**) Histogram depicting the number of genes detected per cell detected by AUCell. (**E**,**F**) AUCell histogram for the Glycolysis gene and Glycolysis model. The glycolysis gene set automatically calculates the activity threshold, and the red line is used to distinguish high/low-activity cell populations. (**G**,**H**) t-SNE plot distinguishing glycolysis High vs. Low activity cells and glycolysis model High vs. Low activity cells detected by AUCell. (**I**,**J**) Heatmap showing the results of CellphoneDB analysis for AUCell glycolysis low-activity cells and high-activity cells, the red box marks the high–activity cells of the glycolysis model in cell communication.

**Figure 7 ijms-26-05105-f007:**
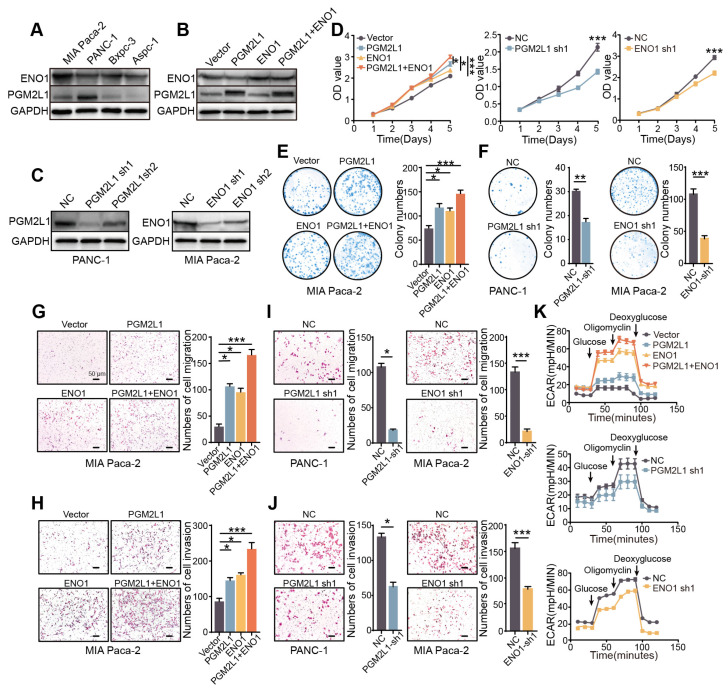
PGM2L1 and ENO1 promote proliferation, migration, invasion, and glycolysis of PAAD. (**A**) Endogenous PGM2L1 and ENO1 expression in four PAAD cells. (**B**) Western blot showing PGM2L1 and ENO1 ectopic expression in transfected MIA Paca-2 cells. (**C**) Western blot showed PGM2L1/ENO1 knockdown by two specific shRNAs. (**D**) CCK8 assay indicating PGM2L1 and ENO1 overexpressed and knocked affected cell growth. (**E**,**F**) Colony formation assay indicated PGM2L1 and ENO1 overexpressed and knocked affected cell colony capacity. (**G**–**J**) Transwell migration and invasion assay showing that PGM2L1 and ENO1 overexpression and knocked affected cell migration and invasion (magnification × 200, scalebar = 50 μm). (**K**) ECAR was measured via Seahorse XF Analyzer in PGM2L1/ENO1-overexpressing or knockdown cells, with glycolytic parameters analyzed in triplicate. Results were summarized as mean ± SD of three independent experiments. (* *p* < 0.05; ** *p* < 0.01; *** *p* < 0.001, independent Student’s *t*-test).

**Figure 8 ijms-26-05105-f008:**
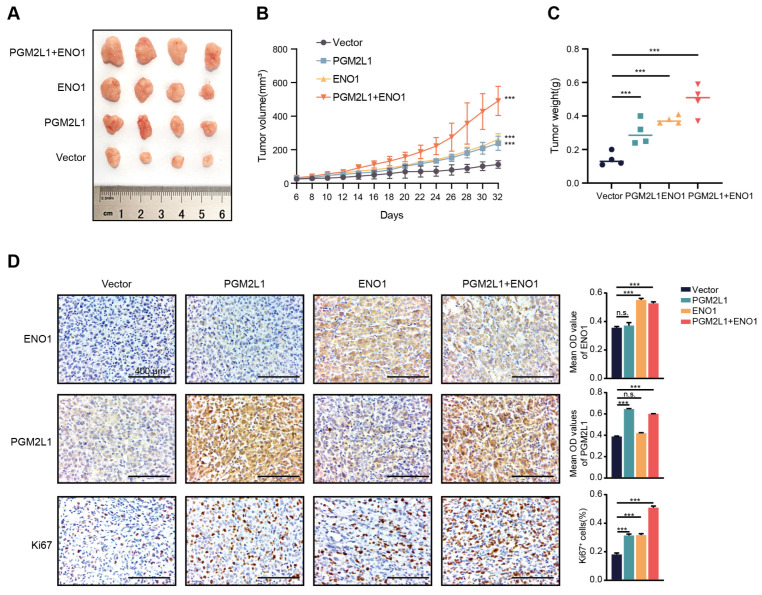
PGM2L1 and ENO1 co-expression xenograft tumor growth in mouse models. (**A**) Xenograft images from subcutaneous injection of transfected MIA Paca-2 cells (vector, PGM2L1, ENO1, PGM2L1+ENO1). (**B**) Tumor volumes were compared among xenografts (*n* = 4) from vector, PGM2L1, ENO1, and PGM2L1+ENO1 groups. (**C**) Tumor weights were compared between vector, PGM2L1, ENO1, and co-expressed groups. (**D**) IHC experiment for PGM2L1, ENO1, and Ki67 in xenograft tumors formed by vector, PGM2L1, ENO1, and co-expressed groups (Scale bars = 400 μm). Results were summarized as mean ± SD of three independent experiments. (*** *p* < 0.001, n.s. = no significance; independent Student’s *t*-test).

## Data Availability

Data are contained within the article, Appendix A, or references cited.

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
