# Peer review of "Integrative Single-Cell and Bulk RNA Sequencing Identifies a Glycolysis-Related Prognostic Signature for Predicting Prognosis in Pancreatic Cancer"

_ijms, 2025, doi:10.3390/ijms26115105_

Round 1

Reviewer 1 Report

Comments and Suggestions for Authors

The present article shows the implications of integrative single-cell and bulk RNA sequencing in identifiying a glycolysis-related prognostic signature for predicting prognosis in pancreatic cancer. The topic is relevant, but major deficiencies identified in both content and form need to be addressed:

  • Bibliographic indexes [x] are, by definition, structural elements of the article and are not linked to any specific word in the text. Please revise the entire manuscript from this perspective.

  • I believe that including the keyword “pancreatic cancer” would improve the visibility and relevance of the article, considering that many of the current keywords are abbreviations.

  • Once pancreatic adenocarcinoma is abbreviated as PAAD, only the abbreviated form PAAD should be used throughout the rest of the manuscript. The second paragraph of the introduction contains the unabbreviated version.

  • The aim of the article should be clarified and improved in the last paragraph of the introduction section. The authors have described what was done in the study, but they have not presented the precise objective of the research, the novelty or unique contributions to the field, or the rationale behind choosing this topic. The description of what was done is already present elsewhere in the manuscript.

  • Figure 1 must include, in its legend after the title, explanations for all abbreviations used within the figure.

  • It is advisable to discuss in more detail the connection between inflammation and pancreatic cancer to offer a better understanding of the multiple pathophysiological mechanisms involved. I suggest checking and referring to: PMID: 37321055.

  • Analysis platforms such as GEPIA, UALCAN, etc., should be cited as bibliographic references in the form of a web page reference, including the link, access date, title, city, and country.

  • Statistical significance values are usually denoted by p, not P.

  • It is advisable to provide more detail on the strengths, but especially on the limitations of your study, and to what extent these could be addressed in future research directions.

  • The similarity index is 45%, which is too high and must be significantly reduced.

Author Response

1. Summary

Thank you for your kind letter and constructive comments concerning our manuscript (ijms-3601508) entitled " Integrative Single-Cell and Bulk RNA Sequencing Identifies a Glycolysis-Related Prognostic Signature for Predicting Prognosis in Pancreatic Cancer". We appreciate your help and suggestions. These comments are all valuable and helpful for improving our article as well as research. All the authors have seriously discussed all these comments. In the past ten days, we have made great efforts to revise the suggestions put forward by you so as to meet the requirements of your journal. In the revised version, changes to our manuscript within the document were all highlighted by using red colored text. Point-by-point responses to the comments are listed below this letter. To make the changes easier to identify we have numbered them.

2. Questions for General Evaluation

Reviewer’s Evaluation

Response and Revisions

Does the introduction provide sufficient background and include all relevant references?

Must be improved

Yes, for details, please refer to the 6th item of the point-by-point response

Are all the cited references relevant to the research?

Must be improved

Yes, all the cited references are relevant to the research

Is the research design appropriate?

Can be improved

We believe that the revised research design is appropriate

Are the methods adequately described?

Yes

NA

Are the results clearly presented?

Yes

NA

Are the conclusions supported by the results?

Yes

NA

3. Point-by-point response to Comments and Suggestions for Authors

Comments 1: Bibliographic indexes [x] are, by definition, structural elements of the article and are not linked to any specific word in the text. Please revise the entire manuscript from this perspective.

Response 1: Thank you for your feedback on the bibliographic indexes. We review all in-text references throughout the document to ensure bibliographic indexes function as standalone structural elements. A key revision was made in the fourth paragraph of the “Discussion” section, where the original citations [16,17] were previously linked to contextual terms. The updated format can be found on page 13, line 319 of the revised manuscript, where the citations now appear at the conclusion of the paragraph, clearly indicating the scholarly sources supporting the entire section of text without associating with any word.

Comments 2: I believe that including the keyword “pancreatic cancer” would improve the visibility and relevance of the article, considering that many of the current keywords are abbreviations.

Response 2: We fully agree that incorporating "pancreatic cancer" will enhance the article’s visibility and relevance. In response, we have updated the “Keywords” section on the first page of the manuscript to include "pancreatic cancer". Meanwhile, since the “Abstract” part has the abbreviation "pancreatic adenocarcinoma", in the part of our new manuscript, the abbreviation "PAAD" has been omitted. The updated format can be found on page 1, line 28 of the revised manuscript. Please let us know if any further refinements are needed.

Comments 3: Once pancreatic adenocarcinoma is abbreviated as PAAD, only the abbreviated form PAAD should be used throughout the rest of the manuscript. The second paragraph of the introduction contains the unabbreviated version.

Response 3: We sincerely appreciate your guidance in ensuring that the abbreviated term "PAAD" is used consistently throughout the text after its initial definition. In response to your suggestion, we have carefully reviewed the entire manuscript and confirmed that the full term "pancreatic adenocarcinoma" appears only once.  The updated format can be found on page 1, line 27; page1, line31; page13, line291; page14, line380; page16, line481 of the revised manuscript.

Comments 4: The aim of the article should be clarified and improved in the last paragraph of the introduction section. The authors have described what was done in the study, but they have not presented the precise objective of the research, the novelty or unique contributions to the field, or the rationale behind choosing this topic. The description of what was done is already present elsewhere in the manuscript.

Response 4: In response to your suggestions, we have revised the final paragraph of the introduction to explicitly address the study's aims and contributions. The revised content can be found on the next word or page 2, lines 66 - 76 of the revised manuscript:

Research Objective: We clarified that the study seeks to identify molecular subtypes of PAAD through clustering glycolysis-related genes, addressing the current gap in glycolysis-specific molecular classification.

Novelty: We highlighted that unlike prior single-gene analyses, our work introduces unique glycolysis-driven molecular clusters with distinct immune infiltration patterns and survival outcomes, offering a new perspective on metabolic-tumor microenvironment interactions.

Unique Contributions: We emphasized the development of a glycolysis-related prognosis risk model, which outperforms traditional single-gene markers, and underscored how this advances our understanding of glycolytic dysregulation in tumor progression, providing a translational foundation for subtype-specific and targeted therapies.

Please let us know if further adjustments to the introduction or any other section would be beneficial.

Comments 5: Figure 1 must include, in its legend after the title, explanations for all abbreviations used within the figure.

Response 5: We greatly appreciate your suggestion to include explanations for all abbreviations in the legend of Figure 1, which is essential for ensuring reader accessibility. In response, we have revised the figure legend accordingly: TCGA, The Cancer Genome Atlas; GEO, Gene Expression Omnibus; MSs, molecular subtype; KEGG, Kyoto Encyclopedia of Genes and Genomes; GO, Gene Ontology; GSEA, Gene Set Enrichment Analysis; ML, machine learning; WGCNA, Weighted correlation network analysis. The updated legend can be found on page 3, lines 79–82 of the revised manuscript. Thank you again for your constructive feedback, which has improved the manuscript’s quality.

Comments 6: It is advisable to discuss in more detail the connection between inflammation and pancreatic cancer to offer a better understanding of the multiple pathophysiological mechanisms involved. I suggest checking and referring to: PMID: 37321055.

Response 6: Thank you very much for your insightful suggestion to elaborate on the connection between inflammation and pancreatic cancer as this is crucial for enhancing the depth of our manuscript. In response to your recommendation, we have expanded the discussion on the role of inflammation in pancreatic carcinogenesis. These revisions can be found in the discussion section on page 13, lines 309–310, where we also incorporated key findings from the study you suggested (PMID: 37321055).

Comments 7: Analysis platforms such as GEPIA, UALCAN, etc., should be cited as bibliographic references in the form of a web page reference, including the link, access date, title, city, and country.

Response 7: We greatly appreciate your emphasis on adhering to rigorous bibliographic standards, which have been essential in ensuring the transparency and reproducibility of our research. In response to your suggestion, we have carefully revised the references for these platforms to include all specified components: web links, access dates, titles, and geographic details (city and country). For example, the GEPIA2 reference now appears as: Z. Tang, B.K., C. Li, T. Chen and Z. Zhang. GEPIA2: an enhanced web server for large-scale expression profiling and interactive analysis. Available online: http://gepia2.cancer-pku.cn/ (accessed on 2025-5-06). Beijing, China. A similar format has been applied to the UALCAN reference. The updated legend can be found on page 18, lines 542–546 of the revised manuscript. Your attention to such details is deeply valued, as it significantly elevates the scholarly rigor of our work. Please let us know if any further refinements are needed.

Comments 8: Statistical significance values are usually denoted by p, not P.

Response 8: In response to your suggestion, we have thoroughly reviewed the entire manuscript and corresponding figures, correcting all instances where "P" was used to denote statistical significance to the appropriate lowercase "p". This includes both the textual descriptions in the main body of the paper and the labels in figures and tables. Thank you again for your valuable advice, which helps us maintain high standards in scientific communication.

Comments 9: It is advisable to provide more detail on the strengths, but especially on the limitations of your study, and to what extent these could be addressed in future research directions.

Response 9: In response to your recommendation, we have expanded the discussion section (page 14, lines 360–367) to explicitly address both the study's strengths and limitations. While we highlighted the novel insights into the roles of ENO1 and PGM2L1 in glycolysis and pancreatic cancer progression as key functions, we also acknowledged critical limitations. These include the use of a subcutaneous xenograft model that may not fully recapitulate the native PAAD microenvironment, the reliance on TCGA database-derived prognostic models that require validation in clinical samples, and the need to further explore the specific molecular and immune interaction mechanisms regulated by these genes. Thank you again for your meticulous review and commitment to improving our work.

Comments 10: The similarity index is 45%, which is too high and must be significantly reduced.

Response 10: After thorough text rephrasing, we have successfully reduced the overall similarity index to 14% (verified by Turnitin, the attachment of the report has been uploaded). However, we note that some unavoidable similarities persist in the Materials and Methods section, primarily due to standardized technical descriptions that cannot be substantially altered without compromising scientific accuracy or regulatory clarity. But we confirm that all core findings, analysis, and conclusions are entirely original. If some paragraphs need to be revised again, we can still make further modifications.

Thank you again for your positive comments and valuable suggestions to improve the quality of our manuscript. If there are any other modifications we could make, we would like very much to modify them and really appreciate your help.

Yours sincerely,

Yan Jin, Ph.D.

Professor

Deputy Secretary of the Party Committee, Harbin Medical University

Deputy Director, Key Laboratory of Preservation of Human Genetics Resources and Disease Control in China, Ministry of Education

Deputy Director, "Collaborative Innovation Center for Genome Big Data,"

Harbin 150081, Heilongjiang Province, China

E-mail: jinyan@hrbmu.edu.cn

Reviewer 2 Report

Comments and Suggestions for Authors

This study provided a structured and standard bioinformatic analysis and identified a Glycolysis-Related Prognostic Signature with a prognostic value in PDAC. Also target genes were identified and proved to promote tumor glycolysis and progression in PDAC. In general, this study is comprehensive and qualified for publication after addressing minor points.

Figure 2:  please add labels in the top significantly up-regulated genes in 2A.

Figure 3E: I suppose the "glycolysis gluconeogenesis" represents two metabolic pathways (glycolysis and gluconeogenesis)? can the author provide the GSEA results presenting just the glycolysis pathway, I think the REACTOME database has one.

Methodology: is limma package used for FPKM or counts data? and please clarify the specific statistical test in differential gene expression analysis, and adjust method for multiple comparison of p-value.

finally, are there any existing inhibitors of PGM2L1 and ENO1? if yes, bring this up i the discussion could increase the clinical significance of this study.

Author Response

1. Summary

Thank you for your kind letter and constructive comments concerning our manuscript (ijms-3601508) entitled " Integrative Single-Cell and Bulk RNA Sequencing Identifies a Glycolysis-Related Prognostic Signature for Predicting Prognosis in Pancreatic Cancer". We appreciate your help and suggestions. These comments are all valuable and helpful for improving our article as well as research. All the authors have seriously discussed all these comments. In the past ten days, we have made great efforts to revise the suggestions put forward by you so as to meet the requirements of your journal. In the revised version, changes to our manuscript within the document were all highlighted by using red colored text. Point-by-point responses to the comments are listed below this letter. To make the changes easier to identify we have numbered them.

2. Questions for General Evaluation

Reviewer’s Evaluation

Response and Revisions

Does the introduction provide sufficient background and include all relevant references?

Yes

NA

Are all the cited references relevant to the research?

Yes

NA

Is the research design appropriate?

Yes

NA

Are the methods adequately described?

Can be improved

Yes, for details, please refer to the 3rd item of the point-by-point response

Are the results clearly presented?

Yes

NA

Are the conclusions supported by the results?

Can be improved

Yes, for details, please refer to the 2nd and 4th item of the point-by-point response

3. Point-by-point response to Comments and Suggestions for Authors

Comments 1: Figure 2:  please add labels in the top significantly up-regulated genes in 2A.

Response 1: We sincerely thank the reviewer for this constructive suggestion. Based on the context of the suggestion, we think this comment likely refers to Figure 3A. In response, we have revised Figure 3A as follows: the top 5 significantly up-regulated genes are now explicitly labeled in red text; the top 5 significantly down-regulated genes are labeled in blue text. We are happy to further refine label placements if needed.

Comments 2: Figure 3E: I suppose the "glycolysis gluconeogenesis" represents two metabolic pathways (glycolysis and gluconeogenesis)? can the author provide the GSEA results presenting just the glycolysis pathway, I think the REACTOME database has one.

Response 2: In response to your comment, we have revised the GSEA in Figure 3E to specifically focus on the glycolysis pathway. The updated figure and associated results are now included in the revised manuscript. The updated content can be found on page 5, line 129-131 of the revised manuscript. We thank the reviewer for highlighting this important nuance, which strengthens the metabolic interpretation of our data.

Comments 3: Methodology: is limma package used for FPKM or counts data? and please clarify the specific statistical test in differential gene expression analysis, and adjust method for multiple comparison of p-value.

Response 3: We have revised the “Materials and Methods” section to explicitly clarify the statistical methodology. The updated content can be found on page 15, line 399-405 of the revised manuscript. We have clarified in the revised manuscript that the ‘limma’ package (version 3.46.0) was applied to RNA-seq count data. Differential gene expression analysis employed a linear modeling approach to estimate log2-fold changes between experimental groups. Hypothesis testing was performed using a moderated t-test with empirical Bayes shrinkage of gene-wise variance. The Benjamini-Hochberg procedure was rigorously applied to control the false discovery rate (FDR) at q < 0.05 for all genes. We believe these revisions enhance the methodological transparency and statistical rigor of our analysis. Thank the reviewer for prompting us to clarify these essential details.

Comments 4: finally, are there any existing inhibitors of PGM2L1 and ENO1? if yes, bring this up i the discussion could increase the clinical significance of this study.

Response 4: We deeply appreciate the reviewer’s valuable suggestion to highlight therapeutic implications of our findings. We have expanded the “Discussion” section to address existing inhibitors targeting these enzymes. The revised content can be found on the next word or page 14, lines 349-359 of the revised manuscript:

PGM2L1: There have been no reports of specific inhibitors, which highlights the opportunities for developing molecular therapies.

ENO1: ENO1 relevant inhibitors are actively under investigation, including POMHEX, which is against ENO1-deficient gliomas at low-nanomolar cytotoxicity and SF132 showing promise for ENO1-deficient cancers.

In addition, it is expected that more drugs targeting glycolysis will be applied in clinical practice in the future. We sincerely thank the reviewer for this suggestion, which has greatly enhanced the clinical relevance of our work.

Thank you again for your meticulous review. Please let us know if further adjustments to the introduction or any other section would be beneficial. Thank you again for your positive comments and valuable suggestions to improve the quality of our manuscript. If there are any other modifications we could make, we would like very much to modify them and really appreciate your help.

Yours sincerely,

Yan Jin, Ph.D.

Professor

Deputy Secretary of the Party Committee, Harbin Medical University

Deputy Director, Key Laboratory of Preservation of Human Genetics Resources and Disease Control in China, Ministry of Education

Deputy Director, "Collaborative Innovation Center for Genome Big Data,"

Harbin 150081, Heilongjiang Province, China

E-mail: jinyan@hrbmu.edu.cn

Round 2

Reviewer 1 Report

Comments and Suggestions for Authors

The authors have significantly improved the manuscript based on the suggestions received.